# Using Temperature Sampling to Effectively Train Robot Learning Policies on Imbalanced Datasets

## Abstract

Increasingly large datasets of robot actions and sensory observations are being collected to train ever-larger neural networks. These datasets are collected based on tasks and while these tasks may be distinct in their descriptions, many involve very similar physical action sequences (e.g., 'pick up an apple' versus 'pick up an orange'). As a result, many datasets of robotic tasks are substantially imbalanced in terms of the physical robotic actions they represent. In this work, we propose a simple sampling strategy for policy training that mitigates this imbalance. Our method requires only a few lines of code to integrate into existing codebases and improves generalization. We evaluate our method in both pre-training small models and fine-tuning large foundational models. Our results show substantial improvements on low-resource tasks compared to prior state-of-the-art methods, achieving better average performance across all tasks under fixed compute budgets. Given fixed model capacity and training time, our method optimally balances the trade-off between high-resource and low-resource task performance, enabling more effective use of model capacity for multitask policies. We also further validate our approach in a real-world setup on a Franka Panda robot arm across a diverse set of tasks.

## 1 Introduction

Scaling robot datasets and model sizes has consistently improved performance on many manipulation tasks (Team et al., 2025; Black et al., 2024). This scaling trend in dataset development emphasizes quantity of data over precise curation of targeted datasets (Dasari et al., 2019; Ebert et al., 2021; Walke et al., 2023; Khazatsky et al., 2024; Collaboration et al., 2023). This development strategy mirrors that of large language models, which also rely on transformer architectures. However, prioritizing scale over content curation produces datasets that under-represent key skills. As a result, models trained on such datasets may develop biased representations, reducing robustness and generalization.

In this work, we focus on learning under imbalanced datasets that mirror the distributions found in recent large-scale robot demonstration collections (Collaboration et al., 2023; Khazatsky et al., 2024). While these datasets exhibit imbalances across multiple dimensions—including language descriptions, camera viewpoints, action primitives, and visual scenes—we argue that action-primitive imbalance deserves particular attention. Unlike vision and language variations, which can be partially addressed through foundation model embeddings (Karamcheti et al., 2024), action-primitive imbalance directly affects the fundamental behavioral distribution of the learned policy. This type of imbalance can lead to models that excel at over-represented actions while failing catastrophically on under-represented but equally important skills, significantly limiting their practical deployment (Dasari et al., 2019; Khazatsky et al., 2024; Ebert et al., 2021; Walke et al., 2023).

Two primary approaches exist for training unbiased models on biased datasets: data augmentation and data reweighting. Given the scale of current robot datasets, data augmentation faces practical limitations. For instance, the Open-X Embodiment Dataset spans

8964.94 GB (Collaboration et al., 2023), so very low resourced skills may require extremely large amounts of generated training data (Garrett et al., 2024). Data reweighting offers a more scalable alternative and has shown promise in robot learning for balancing task representation (Hejna et al., 2024). However, the reweighting in existing works is either based on task complexity (Hejna et al., 2024) or human heuristics (Team et al., 2024) rather than action-primitives. These approaches may still leave fundamental motor skills underrepresented, as complex tasks and primitive actions represent orthogonal dimensions of the learning problem.

**Contributions.** We propose a sampling method for training or fine-tuning robotics policies under data-imbalance which is computationally efficient, simple to implement, and outperforms alternate methods in resulting task success. Under fixed computational budgets and model capacity, our method optimally balances performance across high-resource and low-resource tasks, maximizing average task success. While this may lead to modest performance trade-offs on individual high-resource tasks, we demonstrate that the overall multi-task policy achieves superior performance by better utilizing limited model capacity.

While temperature-based sampling has been explored in multilingual NLP (Wang et al., 2020b; Choi et al., 2023), our work makes several key contributions: (1) we systematically study action-primitive imbalance in robotics, a critical but underexplored dimension distinct from language or vision variations; (2) we demonstrate that robotic policy learning benefits from a *warming* schedule (progressively upsampling low-resource tasks toward the end of training), which is opposite to the *decay* schedules commonly used in NLP; and (3) we provide comprehensive empirical validation across toy domains, simulation (both training from scratch and fine-tuning foundation models), and real-world robotic manipulation. This curriculum-like progression from data-rich to data-scarce tasks mitigates catastrophic forgetting while enabling effective transfer learning, addressing the unique challenges of learning sensorimotor skills under severe data imbalance.

We validate this method on a toy-experimental setup which allows precise control over the task distribution to prove the merits of temperature sampling when training deep learning methods on imbalanced data. We then subsample artificially imbalanced datasets from two simulated robot learning datasets, RoboCasa Nasiriany et al. (2024) and Libero Liu et al. (2023), to validate this approach for robotic policy training in simulation. We demonstrate in simulation that our temperature sampling approach also improves performance for fine-tuning foundation models for robotic policy learning. Finally, we perform real-world experiments with a Franka Panda robot arm using a policy trained from scratch on an imbalanced dataset we collect. We show that in this real-world setting, our new training strategy also increases overall task success.

## 2 TEMPERATURE-BASED SAMPLING FOR IMBALANCED DATA

### 2.1 PROBLEM AND NOTATIONS

We consider the setting of multitask learning on an imbalanced dataset i.e., we have a dataset $D$ containing $m$ tasks of sizes $D = D_1 \cup D_2 \cup \cdots \cup D_m$, where we assume $D_1 >>> D_2 >> D_3 \ldots$. We will refer to tasks with large number of samples as High-Resource Tasks (HRT) and tasks with lower samples as Low-Resource Tasks (LRT). Our aim is to train a policy $\pi_\theta$ with parameters $\theta$ such that we obtain high average performance across the distribution of tasks. For solving a particular task, we sample from our task-conditioned policy, $\pi_\theta(\cdot|x_i, z_i)$ where $x_i$ is the $i^{th}$ input task and $z_i$ is the condition specifying the task; here, a language description embedding of the task.

The behavior cloning objective aims to minimize the negative log-likelihood of expert actions under the task-conditioned policy. Given a dataset of state-action pairs $\{(x_i, a_i, z_i)\}_{i=1}^N$, where $a_i$ is the expert action for input $x_i$ under task $z_i$, the objective is:

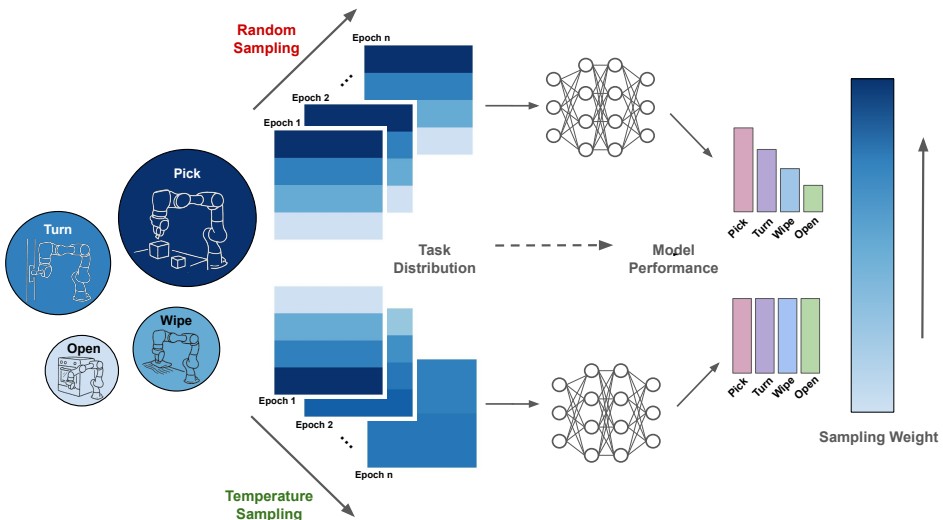

Figure 1: Temperature-based sampling rebalances the distribution over tasks, enabling more equitable training and improved generalization across both high- and low-resource domains.

$$\mathcal{L}(\theta) = -\frac{1}{N}\sum_{i=1}^{N}\log \pi_\theta\left(a_i \mid x_i, z_i\right)$$

This objective encourages the policy to imitate the expert by assigning high probability to the demonstrated actions. To optimize this objective using stochastic gradient descent (SGD), we sample a minibatch $\mathcal{B} \subset D$, and perform the parameter update:

$$\theta \leftarrow \theta - \eta\nabla_\theta\left(\frac{1}{|\mathcal{B}|}\sum_{(x_i, a_i, z_i)\in\mathcal{B}} -\log\pi_\theta\left(a_i \mid x_i, z_i\right)\right)$$

where $\eta$ is the learning rate. In practice, the composition of $\mathcal{B}$ can significantly influence the learning dynamics in the presence of task imbalance, since tasks with more data dominate the gradient updates throughout the training.

## 2.2 Temperature-Based Sampling

To mitigate representation bias due to data imbalance across tasks, we employ temperature-based sampling, a strategy that reshapes the sampling probabilities over tasks to up-sample low-resource ones and downsample high-resource ones. Given task dataset sizes $\{|D_1|, |D_2|, \ldots, |D_m|\}$, the sampling probability for task $i$ under temperature $\tau > 0$ is:
$p_i^{(\tau)} = \frac{|D_i|^{1/\tau}}{\sum_{j=1}^{m}|D_j|^{1/\tau}}$

This temperature $\tau$ acts as a knob controlling task balance:

- $\tau = 1$ : recovers sampling proportional to dataset size (referred to as random sampling method throughout the text; as this falls back to the most commonly used random sampling strategy for training deep neural networks).

- $\tau > 1$ : increases the relative probability of smaller (low-resource) tasks.

- $\tau < 1$ : further biases sampling toward high-resource tasks.

This formulation is analogous to a Boltzmann distribution, where $\log |D_i|$ represents the "energy" of each task and $\tau$ acts as the temperature. Unless otherwise mentioned, we follow a cosine warming temperature schedule over the training epochs formulated as, $T_{\text{warmup}}(t) = T_{\text{start}} + (T_{\text{end}} - T_{\text{start}}) \cdot \frac{1 - \cos(\pi t)}{2}$, with initial temperature of 1 and end temperature of 5, i.e., we upsample low-resource tasks near the end. We choose this temperature range and schedule after a thorough hyper-parameter search, where cosine warming outperformed cosine decay, both linear and exponential warmup, and both linear and exponential decay.

For one of the baselines we also use fixed temperature for upsampling low-resource tasks, unless otherwise mentioned the temperature will be $T = 5$, which is chosen from among the most-commonly used upsampling rates of 2, 3.3, 5 based on initial experiments. Random sampling, which is most widely used can also be viewed as a special cases of temperature sampling where $T = 1$.

## 3 Experiments

To prove the efficacy of this data sampling method, we evaluate it in three different settings. Firstly, we show on a toy-experimental setup which allows precise control over the task distribution to prove the merits of temperature-based sampling in the presence of data-imbalance. After that, we create an imbalanced simulated data subsetted from Robocasa and Libero to prove the same in robotic policy training from scratch as well as fine-tuning foundational models. And finally, we take a real-world imbalanced dataset to concretize the efficacy of our method in real-world scenarios.

### 3.1 Toy Experiment: Sparse Parity

We use sparse parity task as a minimal, controlled multitask setting to study the impact of temperature-based data sampling. Borrowed from computational learning theory, sparse parity has been a long-standing benchmark for analyzing algorithm performance and feature selection. Each task corresponds to predicting the parity (even or odd) of a specific subset of input bits in a binary vector. Importantly, the subsets used by different tasks do not overlap, ensuring clear task boundaries and no shared information across tasks. Although this separation may not reflect real-world datasets where positive transfer between skills often occurs, it provides a clean experimental environment for isolating the effects of sampling strategies by controlling the exact task-distribution. This setup allows us to independently vary the frequency of each task, making it well-suited for analyzing how temperature scheduling affects learning dynamics in imbalanced multitask scenarios.

In a deep-learning setup, sparse parity task requires a model to compute the parity (sum modulo 2) of a subset of bits from a binary input string. In our experiments, we use input strings of length n=50 and randomly select k=4 positions for a task. We extend this to a multi-task learning scenario with five different parity tasks, where the model receives both a one-hot encoded task identifier and the binary input string, and must learn to compute the correct parity function for each task. The frequency of tasks for training follows a power law distribution controlled by two hyperparameters, allowing us to simulate realistic scenarios where some tasks appear more frequently than others. A visual description of the task and further details are given in Appendix A. Fig. 2 shows the faster convergence over the tasks as we over-sample low-resource tasks using temperature based control. Although in this

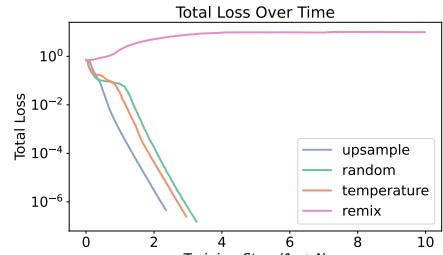

Figure 2: Total training loss over time for sparse parity tasks shows improvement in performance by over-sampling low-resource tasks in imbalanced dataset. While also highlighting the limitations of adaptive sampling methods like ReMix due to sensitivity to hyper-parameters.

environment, the loss benefits from higher upsam-
pling temperature rather than annealing process
like our proposed schedule in real-world environments this usually leads to overfitting. We
notice that ReMix (Hejna et al., 2024) fails to converge due to extreme sensitivity to hyper-
parameters like early-stopping and rapidly changing weights due to adaptive weights based
on loss, highlight the main limitations of adaptive sampling techniques.

## 3.2 ROBOTIC MANIPULATION SIMULATION EXPERIMENTS

**Dataset:** For evaluating our hypothesis in robot policy learning, we use two of the most
widely used simulation datasets in robotics, RoboCasa Nasiriany et al. (2024) and Libero
Liu et al. (2023).

Robocasa provides a large-scale dataset of 25 atomic tasks grounded in 8 foundational sen-
sorimotor skills, such as pick-and-place, door and drawer manipulation, pressing buttons,
and turning knobs. Each atomic task is accompanied by 50 human demonstrations and 3000
synthetic demonstrations, totaling 1,200 human-collected trajectories and 72000 synthetic
demonstrations. The tasks are embedded within 120 visually diverse kitchen scenes, featur-
ing randomized layouts, styles, and AI-generated textures, offering rich perceptual and in-
teraction diversity. While Libero provides 4-suite of tasks, Libero-SPATIAL, Libero-GOAL,
Libero-OBJECT, & Libero-10. Each task-suite contains 10 tasks, and are accompanied by
50 human-demonstrations for each task. More details about the tasks are in Appendix C.

To simulate data imbalance, we construct a skewed distribution by selecting 3,000
demonstrations for seven pick-and-place tasks, while retaining only 50 demonstrations for
each of the remaining atomic tasks in Robocasa. For Libero, we use 50 demonstrations for
Libero-SPATIAL, 15 for Libero-OBJECT and GOAL, and 20 for Libero-10. We construct
these skewed distributions to represent common characteristics of current imbalanced robot
learning datasets which also align with real world imbalances. Specifically, pick-and-place
tasks are by far the most common in our skewed sampling of Robocasa, and spatial tasks
which vary the spatial configurations of repeating objects and goals are represented more in
our skewed sampling of Libero.

**Baselines:** We compare our sampling with 3 baselines. First, we use random sampling
which represents picking the datapoints randomly from the given dataset. Random sam-
pling is one of the most common methods used in robotic pre-training. We also upsample
tasks based on their size in the training dataset, i.e., less common tasks are sampled more
frequently throughout training. We also benchmark ReMix Hejna et al. (2024), which uses
a group-distributional robust optimization method to balance datasets. In contrast to our
method and other baselines, the weights for the datasets/tasks are not dependent on the
size but rather on the difficulty which is measured by training a reference model and a proxy
model and measuring relative loss of the datasets/tasks on these models to assign weights
for final model training. Hence, this method requires training two additional models before
training the target policy. ReMix is therefore sensitive to hyper-parameters such as model
sizes and gradients steps used for estimating weights.

**Training and Evaluation:** We train a single multi-task policy using the Behavior Cloning-
Transformer (BC-T) Nasiriany et al. (2024), with 512 embedding dimensions, 6 layers, and
8 attention heads on our imbalanced Robocasa dataset. On our imbalanced Libero dataset,
we fine-tune UniVLA, a robotics foundation model pretrained on human-videos (Ego4D
Grauman et al. (2021)) and large-scale robotic datasets (Cross-X embodiment Collaboration
et al. (2023) & Bride-V2 Walke et al. (2023)) on all the Libero-tasks. In both cases, we
perform 40k gradient steps; more details about hyper-parameters are in Appendix C.

## 3.3 ROBOTIC MANIPULATION HARDWARE EXPERIMENTS

**Dataset:** In the real-world, we perform table-top manipulation experiments with a Franka
Panda Emika 7-DoF robot arm, setup as shown in Fig. 3a. We train a diffusion policy with
a UNet-based architecture and a ResNet-50 visual encoder from scratch on an imbalanced
dataset consisting of 8 tasks with a total of 588 demonstrations. The exact distribution of

the tasks is given in Fig. 3b. This policy is trained with RGB observations from three cameras—two wrist-mounted and one egocentric—along with proprioceptive state and language embeddings.

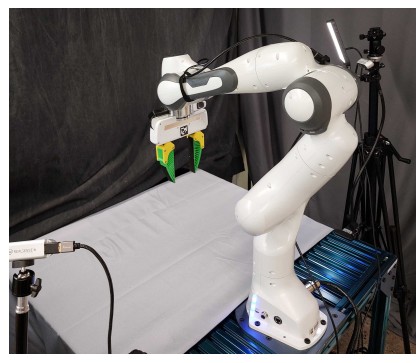

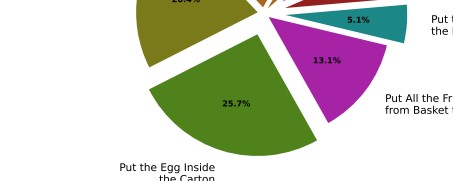

(a) Setup for the real world Experiments.

(b) Tasks distribution for the real world experiments. With highlighted tasks chosen for evaluation

Figure 3: Real-world experiment setup as well as task distributions.

**Training and Evaluation:** We train a single multi-task, CNN-based Diffusion Policy Chi et al. (2024) for 40k gradient steps and evaluate performance on four tasks (one high-resource and three low-resource tasks): 1) *Egg in Carton* (a highly represented pick-and-place task), 2) *Pen in Mug* (a less represented pick-and-place task), 3) *Fold Towel* (a less represented folding task), and 4) *Slide Cube* (a less represented sliding task). Each policy is rolled out for 10 trials per task, and we report the average success rate. Further training details are provided in Appendix B.

## 4 RESULTS & DISCUSSION

### 4.1 DOES TEMPERATURE-BASED SAMPLING IMPROVE THE PERFORMANCE ON LOW-RESOURCE TASKS?

Does temperature-based sampling of training data improve robotic policy performance compared to naively sampling, basic upsampling, or sophisticated distributional weighing schemes in the presence of data-imbalance? To answer this question, we look at our evaluations across training and fine-tuning on three different datasets in both simulation and real-world environments. Fig. 4, Table 1 and Fig. 5 show that temperature sampling allows an absolute increase in policy performance compared to other methods. Most of the improvement comes from low-resource tasks, but when comparing to constantly upsampling low-resource tasks, temperature-sampling allows for better utilization of the model capacity. This highlights the importance of an annealing process, which we hypothesize leads to diverse gradients throughout training and thus better generalization. Specifically, we believe the cosine warming schedule (upsampling low-resource tasks toward the end of training) mitigates catastrophic forgetting: by first allowing the model to learn robust representations from abundant high-resource data, then progressively emphasizing low-resource tasks, the policy can specialize on underrepresented skills without completely overwriting previously learned behaviors. This curriculum-like progression from data-rich to data-scarce tasks enables the model to leverage transfer learning while maintaining performance across all tasks.

### 4.2 ABLATION STUDIES:

We perform ablation studies on Robocasa dataset to evaluate robustness of our method across varying model sizes, imbalance distribution, and schedules.

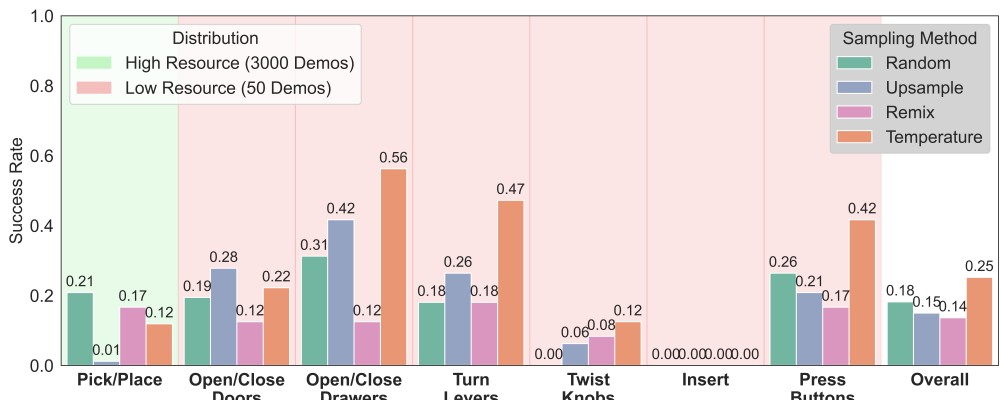

Figure 4: Average subtask success rates in simulation for high- and low-resource tasks under different sampling strategies. Temperature sampling provides the highest performance on low-resource tasks and achieves the best average performance across all tasks under fixed compute budget, optimally balancing the capacity trade-off between high- and low-resource tasks.

|  | Libero Spatial | Libero Goal | Libero Object | Libero 10 | Overall |
|---|---|---|---|---|---|
| Random | 0.90 | 0.74 | 0.80 | 0.60 | 0.76 |
| Upsample | **0.96** | 0.63 | 0.80 | 0.68 | 0.77 |
| Temperature | **0.96** | **0.86** | **0.84** | **0.73** | **0.85** |

Table 1: Fine-tuning on UniVLA on an imbalanced Libero-dataset with data ratio of {1.0, 0.3, 0.3, 0.4} for Libero Spatial, Goal, Object and Libero-10 respectively. Temperature sampling outperforms randomly sampling or fixed upsampling without hurting the performance on high-resource tasks.

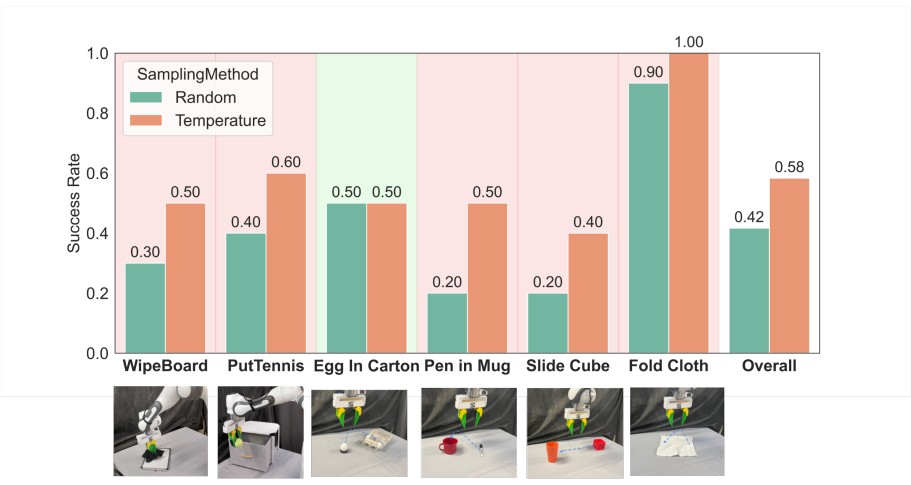

Figure 5: Real-world performance on a Franka Panda robot across four tasks, comparing random, cosine decay, and cosine warming schedules. Cosine warming consistently improves success rates, particularly for low-resource tasks.

To better understand the mechanism for better performance of temperature sampling and robustness of the method under varied situations, we design a series of ablation studies with Robocasa dataset.

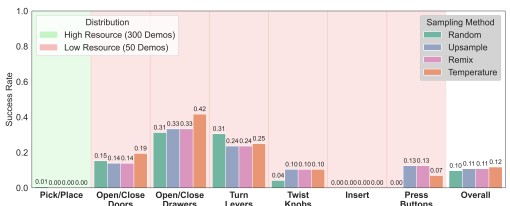

(a) 300:50 data imbalance distribution

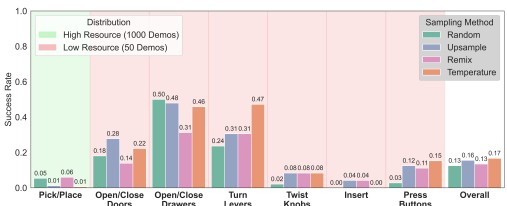

(b) 1000:50 data imbalance distribution

Figure 7: Average subtask success rates in simulation for high- and low-resource tasks under different sampling strategies. Results are shown for two data imbalance distributions: 300:50 (left) and 1000:50 (right), representing the number of demonstrations for pick-and-place versus other tasks. Cosine warming provides the highest performance on low-resource tasks and achieves the best average performance across all tasks, optimally balancing the capacity trade-off between high- and low-resource tasks under fixed compute constraints.

**How does the performance vary with model sizes?** Transformer models have a fixed capacity that scales with their model size. We would like to understand how different sampling methods affect the use of these model capacities. Fig. 6 shows different sampling methods with transformer sizes of Small (3.1M), Base (19M), Large (56.7M).

We observe that when having limited capacity, randomly sampling datapoints leads to underutilization of the model capacity and the model fails to generalize to any tasks. In such cases, using temperature sampling allows for efficient usage of capacity leading to stronger models at smaller scales, while also benefiting from increased model scale.

**How does performance vary with different imbalance ratios?** To understand how different sampling methods perform under different imbalance ratios, we evaluate them on two more imbalance ratios (1000:50, 300:50) in addition to our previous imbalance ratio (3000:50). Fig. 7 shows that temperature sampling helps in low-resource tasks over our random baseline, and the gap between performance of baselines and temperature sampling increases as the imbalance ratio increases, highlight the need for temperature-sampling methods as imbalance increases. We note that the absolute performance on pick-and-place tasks in these extreme imbalance settings is low across all methods, consistent with findings in the original RoboCasa work (Nasiriany et al., 2024). This is due to the high task diversity (dozens of object categories with varying affordances) and dexterity requirements that make pick-and-place particularly challenging compared to tasks like door manipulation with fewer object variations. The relative improvements between methods remain meaningful for comparing sampling strategies.

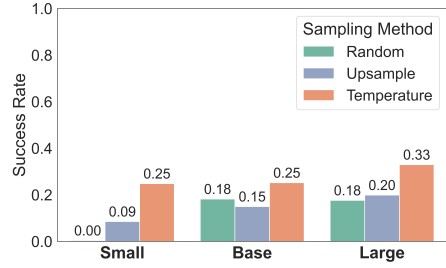

Figure 6: Impact of temperature-based sampling across different model sizes in simulation. Cosine warming maintains its advantage in both smaller and larger transformer architectures, indicating robustness across model capacity regimes.

**Does schedule matter over annealing process?** We ask whether the reason behind the success of temperature-based scheduling is the cosine schedule used in our experiments or the temperature warming schedule (T=1 to T=5 over training). To answer this question, we try two additional schedules (linear and exponential) with temperature warming (T=1 to T=5) and temperature decay (T=5 to T=1). Table. 2 shows that while sampling the low-resource tasks more at the end (warming) tends to perform better on low-resource tasks, however, the type of schedule also effect the model's final performance. Identifying the reasons for such difference in performance for different schedules is an important direction for future work.

**Summary:** We study multi-task policy learning under severe task imbalance and propose temperature-based sampling to train robust multi-task imitation learning policies for robotic manipulation. Across toy domains, simulation, and real-world robotics, our method substantially improves performance on low-resource tasks and achieves better average performance across all tasks under fixed compute constraints. While individual high-resource tasks may experience modest performance reductions due to limited model capacity, the overall multi-task policy demonstrates superior generalization. Ablations show that gains hold across model sizes for a fixed schedule, but depend on the schedule shape, with cosine schedules performing best. Moreover, the benefit of temperature-based sampling grows with the imbalance ratio, underscoring the need to better understand temperature scheduling for training multi-task robot learning policies under skewed task distributions.

|  |  | Warming | Decay |
|---|---|---|---|
| **High-resource** | Cosine | 0.12 | **0.17** |
|  | Exponential | 0.02 | **0.04** |
|  | Linear | 0.02 | 0.02 |
| **Low-resource** | Cosine | **0.31** | 0.24 |
|  | Exponential | **0.28** | 0.24 |
|  | Linear | **0.24** | 0.16 |

Table 2: Comparison of task success rates across scheduling strategies by resource regime. Warming schedules which sample low-resource tasks higher during the end of training allow for better generalization on low-resource tasks.

## 5 RELATED WORK

**Training on Imbalanced Datasets.** Training unbiased models on biased datasets is a widespread problem for machine learning practitioners. The two most common methods for mitigating bias in models trained on biased datasets are data augmentation and data reweighing. Data augmentation artificially enhances minority class representations. To perform data augmentation on real world robotic datasets, generative models have been used to generate additional photorealistic trajectories (Mandi et al., 2022; Chen et al., 2023; Kapelyukh et al., 2023; Yu et al., 2023). This generated data is shown to be effective in improving visuomotor policy performance. Data reweighing modifies the training process by assigning higher weights to under-represented examples or tasks. Recent work in visuomotor policy learning for robotics reweighs domains partitioned by estimated task complexity (Hejna et al., 2024). This reweighing is performed through an estimate of excess loss given by a trained reference model. In contrast, our work proposes to address data imbalance in represented action sequences. We reweigh domains via simply ordering data presentation to the model during training, more specifically this is usually referred to as upsampling the dataset and is proven to induce lower variance gradients than reweighing (Li et al., 2024). We also experiment on the order which matters for different classes of models since the timing of the introduction of low resourced data did not yield the same result in different network architectures in natural language processing (Choi et al., 2023) and computer vision applications (Li et al., 2024).

**Curriculum Learning.** In multi-task and multi-lingual learning settings, which suffer from similar problems of highly-imbalanced datasets, curriculum learning has been widely studied to mitigate bias incurred from data imbalance (Jean et al., 2019; Wang et al., 2020a; Kreutzer et al., 2021; Wang et al., 2020b; Choi et al., 2023).Notably, Wang et al. (2020b), Choi et al. (2023) propose similar temperature-based scheduling approach as ours, concluding very similar results as ours. Temperature sampling (Wang et al., 2020b; Choi et al., 2023) provides a simple and efficient way to address imbalance in contrast to more adaptive methods (Jean et al., 2019; Wang et al., 2020a; Kreutzer et al., 2021). This prior work primarily evaluates temperature sampling approaches on multi-lingual datasets. In contrast, our contribution is in robot learning, addressing an analogous problem in a different domain, requiring a reformulation of the sampling problem and distinct evaluation. In particular, our work proposes performing sampling over imbalanced primitive actions and requires evaluation using a physical robot arm to validate impact on final task execution.

**Skill Decomposition.** Our domains are partitioned based on estimates of atomic skills. Segmenting robot trajectories into atomic skills or primitives has been explored extensively, with existing approaches falling into three main categories: predefined motion primitives (Kober & Peters, 2009; Niekum et al., 2012; Peters et al., 2013), contact mode-based segmentation (Toussaint et al., 2018; Su et al., 2018; Silver et al., 2022), and latent representation methods (Shankar et al., 2020; Kipf et al., 2018; Nasiriany et al., 2024; Zhang et al., 2024; Memmel et al., 2024; Lin et al., 2024). Each approach presents distinct trade-offs. Predefined primitives limit generalizability to new tasks, while contact-based methods require additional sensory data that is often missing from modern data collection pipelines. In contrast, latent representation methods offer an attractive balance by flexibly encoding both semantic and non-semantic characteristics needed by downstream tasks Nasiriany et al. (2024); Zhang et al. (2024); Memmel et al. (2024); Lin et al. (2024). While these alternative segmentation methods should theoretically yield similar policy training results with our temperature sampling approach, experimental validation is beyond the scope of this paper. For our experiments, we assume clear task segmentation in our dataset based on skills.

## 6 LIMITATIONS & FUTURE WORK

In this paper, we propose a sampling method for training or fine-tuning robotics policies under data-imbalance which is computationally efficient, simple to implement, and outperforms alternate methods in results task success. We validate this method on a toy experimental setup, simulated robot manipulation, and real world robot manipulation. We find across all these settings that temperature-based sampling with cosine-warming achieves the overall best task performance in scenarios where the target dataset is imbalanced.

**Limitations.** First, we applied temperature-based sampling with a fixed schedule which requires determining and setting the training steps before launching the run. Further work should investigate schedules which are less sensitive to training steps and allow for continual learning of tasks. Furthermore, a significant limitation of our work is the assumption of segmented tasks and knowing their frequency prior to training. This assumption hinders adaptability of our method to wider training datasets with heterogeneous skills and no clear segmentation between them. However, in our related work, we review the extensive research on learning segmentations of these skills in unstructured datasets. This research direction provides a promising path toward enabling automated labeling of data into decomposed skills, allowing our approach to apply to any robotic dataset.

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

# A   TASK DESCRIPTION AND TRAINING DETAILS FOR SPARSE PARITY

**Task Description:** We evaluate our sampling strategies on the *multi-task sparse parity task*, a synthetic benchmark where each task requires computing the parity (XOR) over a task-specific subset of $k = 3$ bits from a binary input vector of length $n = 50$. We construct $T = 10$ tasks using *random task selection*, where each task independently samples a distinct subset of 3 input bits. To enable task-conditioned learning, each input is augmented with a one-hot task identifier of dimension $T$, resulting in an input vector of dimension $n + T = 60$. Fig. 8 provides an example for the case when n = 3, T = 10.

Training data is drawn according to a *Zipfian distribution* to introduce imbalance across tasks. Specifically, the probability of sampling task $t \in \{1, \ldots, T\}$ is given by

$$P(t) = (t)^{-\alpha} \bigg/ \sum_{i=1}^{T} (i)^{-\alpha},$$

where $\alpha = 1.5$. This results in a heavy-tailed distribution where earlier-indexed tasks are sampled significantly more often.

**Training Details:** We use a two-layer multilayer perceptron (MLP) with 100 hidden units per layer and ReLU activations. The model is trained using the Adam optimizer for 250,000 steps with a batch size of 10,000, learning rate of 0.001, and no weight decay. Evaluation is performed on a balanced test set of 10,000 samples (1,000 per task). This setup provides a controlled testbed for analyzing the effects of task imbalance and sampling policies.

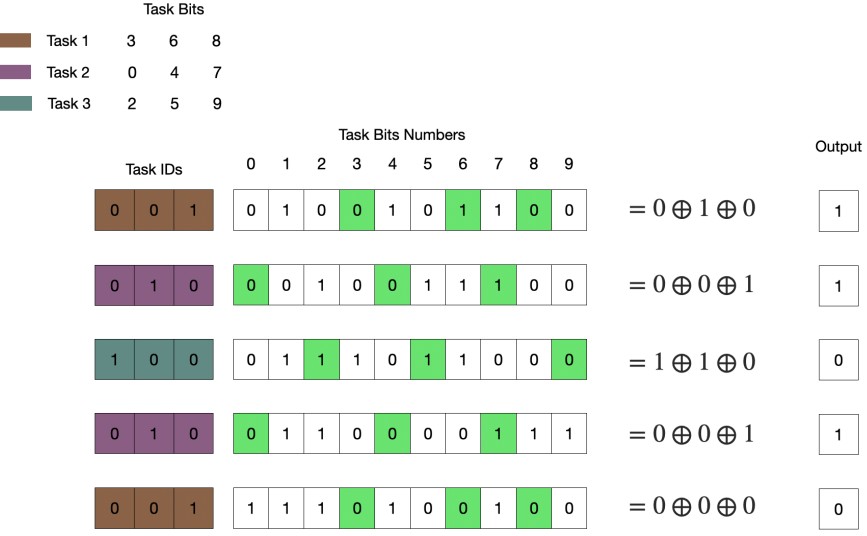

Figure 8: A three-task sparse parity task with the parity bits for the tasks chosen at random (top left). The figure shows five examples (Task Bits Numbers) for different tasks, which are input to the network along with the one-hot Task IDs (colored column), and their corresponding outputs based on the parity of the parity bits.

# B  Task Descriptions and Training Details for Simulation Experiments

**Task Details**

**Robocasa**: We construct a dataset using RoboCasa's atomic tasks, which encompass eight foundational sensorimotor skills including pick-and-place, opening and closing doors and drawers, pressing buttons, twisting knobs, and other basic manipulations. To simulate data imbalance, we curate a subset from RoboCasa consisting of 3,000 demonstrations from seven pick-and-place tasks, and only 50 demonstrations each for the remaining atomic tasks. This synthetic imbalance mirrors the skill distribution biases commonly found in large-scale robotic datasets. Demonstrations consist of both high-quality human teleoperation data and large-scale synthetic data generated via the MimicGen system, as described in Nasiriany et al.,Nasiriany et al. (2024).

**LIBERO** A lifelong robot manipulation benchmark built via a procedural pipeline that generates language-conditioned tasks (instructions, PDDL initial states, and goal predicates), providing 130 standardized tasks for studying architectures, algorithms, and pretraining.

- **LIBERO-SPATIAL (10 tasks).** Same objects/goals; two identical bowls differ only by location/relations—tests continual acquisition of spatial-relational knowledge.
- **LIBERO-OBJECT (10 tasks).** Same layout/goals; each task changes the target object—tests continual learning of new object concepts for pick-and-place.
- **LIBERO-GOAL (10 tasks).** Same objects with fixed spatial configuration; only the task goal changes—tests continual learning of new motions/behaviors.
- **LIBERO-10 (long-horizon).** A 10-task subset of LIBERO-100 containing only long-horizon tasks; used for downstream evaluation after pretraining on the short-horizon split.

**Training Details:** We train Behavior Cloning-Transformer (BC-T) similar to Nasiriany et al. (2024), with an observation history of 10 and action prediction horizon of 10 and executing 1 action before replanning. We optimize our policy network using the AdamW optimizer with a weight decay regularization coefficient of 0.01. The initial learning rate is set to 1e-4, and we employ a constant learning rate schedule with a warm-up phase lasting for the first 100 epochs. This schedule ensures stable initial training dynamics before transitioning into the main training phase. For the objective function, we use a weighted combination of loss terms with an L2 loss coefficient of 1.0. We train our policy for 40,000 gradient steps with a batch size of 16.

# C  Task Descriptions and Training Details for Real-World Experiments

**Task Setup and Distribution:** For the real-world experiments, we collect an imbalanced data set consisting of a varying number of demonstrations for 8 tasks with a total of 588 demonstrations on a Franka Panda Emika 7-DoF arm.

**Training Details:** We train a Diffusion Policy model with a UNet-based architecture and a ResNet-50 visual encoder using RGB observations from three cameras—two wrist-mounted and one egocentric—along with proprioceptive state and language embeddings. Images are resized to $128 \times 128$ and augmented with color and crop randomization.

The UNet consists of three levels with downsampling dimensions $[256, 512, 1024]$, a kernel size of 5, and group normalization. It predicts 8-step action sequences over a 16-step prediction horizon, conditioned on a single observation. Training is conducted for 50,000 gradient steps with a batch size of 128, using the AdamW optimizer with an initial learning rate of $1 \times 10^{-5}$, no weight decay, and a constant learning rate schedule. We train with 100 denoising steps and use DDIM inference with 10 sampling steps and 8 noise samples per training example.

