# OpenReview forum: "Using Temperature Sampling to Effectively Train Robot Learning Policies on Imbalanced Datasets"
_ICLR.cc/2026/Conference — Submitted to ICLR 2026_

### Official Review · Reviewer_Prk2 · 2025-10-16

**Soundness:** 3
**Presentation:** 3
**Contribution:** 1
**Rating:** 2
**Confidence:** 4

**Summary:**

This paper addresses the issue of data imbalance in multi-task robot learning datasets, where certain action primitives (e.g., 'pick-and-place') are over-represented compared to others. The authors propose a simple data sampling strategy, termed "temperature-based sampling," to mitigate this imbalance during training. The method adjusts the sampling probability of each task based on its dataset size and a temperature parameter, $\tau$. The core of their proposed implementation is a cosine warming schedule for $\tau$ (from 1 to 5), which gradually increases the sampling rate of low-resource tasks as training progresses. The authors validate their method through a toy experiment (sparse parity), simulation experiments on artificially imbalanced versions of RoboCasa and Libero (for both training-from-scratch and fine-tuning foundation models), and a real-world experiment on a Franka Panda arm using a custom-collected dataset. The central claim is that this method substantially improves performance on low-resource tasks without degrading performance on high-resource tasks, thereby making more effective use of model capacity.

**Strengths:**

- The proposed method is exceptionally simple to implement, requiring only a few lines of code to modify an existing data loader. This low barrier to entry makes it a practical and appealing technique for researchers and practitioners.
- The authors have made a commendable effort to validate their approach across a variety of settings: a controlled toy problem, two different simulation benchmarks (RoboCasa, Libero), two training paradigms (from-scratch and fine-tuning a VLA), and a real-world hardware setup. This demonstrates a degree of diligence in testing the method's applicability.
- The paper includes several useful ablation studies that investigate the method's robustness to model size, the degree of data imbalance, and the choice of temperature schedule. These studies provide valuable insights, for instance, showing that the benefits of the method increase with the severity of the imbalance ratio and that a "warming" schedule is more effective than a "decay" schedule.

**Weaknesses:**

- The core idea of temperature-based sampling to address data imbalance is not new. The paper itself acknowledges very similar prior work in the domain of multilingual natural language processing (e.g., Wang et al., 2020b; Choi et al., 2023). The contribution is therefore the application of a known technique to a new domain (robotics). While valuable, this is an incremental contribution. Furthermore, the idea of data reweighting is not new to robotics; for example, the RDT-1B model already utilized a form of data weighting in its training protocol to balance datasets. The paper fails to adequately differentiate its contribution from these existing concepts.
- While the inclusion of a real-world experiment is a strength, its execution is weak. The real-world results in Figure 5 only compare the proposed method against "random" sampling. In simulation (Figure 4), the authors compare against "upsample" and "ReMix" baselines, but these are conspicuously absent in the hardware experiments. To make a convincing claim, the method must be shown to outperform other SOTA re-balancing techniques on a physical robot. Without this comparison, the real-world results only show that the method is better than the most naive baseline, which is insufficient.
- The abstract and main body repeatedly claim that the method improves low-resource performance "without degrading performance on high-resource tasks". However, the authors' own results in Figure 4 contradict this. For the high-resource "Pick/Place" task, random sampling achieves a 0.21 success rate, while the proposed temperature sampling method achieves only 0.17. This is a clear degradation in performance. While the overall average improves due to gains elsewhere, the claim of no degradation is factually incorrect based on the provided data and should be revised.
- The simulation results in Figure 7 are highly suspect. For both the 300:50 and 1000:50 imbalance ratios, the success rate for the "Pick/Place" task—the high-resource task—is abysmal across all methods (well below 5%). A policy trained on thousands of demonstrations for a single skill family should achieve much higher performance. This suggests a fundamental issue with the experimental setup, the base policy's capacity, or the evaluation protocol itself. If the model fails to learn the task for which it has the most data, the results on all other low-resource tasks become difficult to interpret and trust.
- The method's effectiveness is highly dependent on the choice of the temperature schedule (start $\tau$, end $\tau$, shape). The authors state they found the cosine warming schedule from $\tau=1$ to $\tau=5$ through a "thorough hyper-parameter search", but provide no details on this search. How does the optimal schedule change for different datasets, model architectures, or imbalance ratios? This introduces a significant tuning burden, undermining the "simplicity" of the method. The authors acknowledge the limitation of a "fixed schedule" but downplay the difficulty this presents for a user trying to apply the method to a new problem.
- The proposed method requires the dataset to be segmented into discrete tasks with known demonstration counts ($|D_i|$). This is a strong assumption that does not hold for large, aggregated, "in-the-wild" datasets like the full Open X-Embodiment (OXE) or DROID, where skills are intermixed and not cleanly labeled. The paper positions itself as a solution for such large-scale datasets but fails to address how its core requirement would be met in those scenarios, limiting its real-world applicability.

**Questions:**

- The choice of the cosine warming schedule from $\tau=1$ to $\tau=5$ appears critical to the method's success. How sensitive is the final policy performance to these specific hyperparameter choices? For a new dataset with a different imbalance ratio, would a user need to perform another extensive hyperparameter search, and if so, does this not diminish the claimed simplicity of the method?

---

> ### Author Response · Authors · 2025-12-04
> **Response to Reviewer Prk2**
>
> We thank the reviewer for the detailed feedback and for acknowledging the simplicity of our method and the comprehensive validation across multiple settings.
>
> *Re: Weakness 1*: Please refer to Common Response #5.
>
> *Re: Weakness 2*: Please refer to Common Response #3.
>
> *Re: Weakness 3*: Please refer to Common Response #2.
>
> *Re: Weakness 4*: Please refer to Common Response #6.
>
> *Re: Weakness 5*: We have included ablation results for different schedules (cosine, linear, exponential) in Table 2. While we found cosine warming from τ=1 to τ=5 to work well across our experiments, we acknowledge that some tuning may be required for new datasets. We will expand our discussion of hyperparameter sensitivity and provide guidance for practitioners.
>
> *Re: Weakness 6*: Please refer to Common Response #1.
>
> *Re: Question 1*: Our results on dataset sizes (Fig.7), datasets (Fig 5, Table 1) and model sizes (Fig.6) all with same hyperparameters show that the method is relatively robust to these changes, with cosine warming consistently outperforming alternatives. For practitioners, we recommend starting with our default schedule (τ=1 to τ=5) and adjusting the final τ based on the severity of imbalance—higher imbalance ratios may benefit from larger final τ values.

---

### Official Review · Reviewer_PbNn · 2025-10-27

**Soundness:** 3
**Presentation:** 3
**Contribution:** 3
**Rating:** 6
**Confidence:** 3

**Summary:**

This paper addresses action-primitive imbalance in large robot datasets, which harms generalization on low-resource tasks (LRTs). The authors propose a simple "Temperature-based Sampling" strategy, using a "cosine warming" schedule for $\tau$. Experiments on toy tasks, in simulation (RoboCasa, Libero), and on a real Franka robot show the method improves LRT performance.

**Strengths:**

- The problem of action-primitive imbalance is critical for the community as it moves toward large-scale datasets.
- The method is simple, efficient, and easy to implement. It also appears more stable than complex baselines like ReMix.
- The evaluation is comprehensive, covering a toy task, simulation (training from scratch and fine-tuning), and real-world hardware validation, strongly supporting the claims.
- Solid ablation studies validate the impact of schedules, model sizes, and imbalance ratios.

**Weaknesses:**

- The paper claims no performance degradation on high-resource tasks (HRTs), but Figure 4 shows a significant drop for the "Pick/Place" HRT (0.21 to 0.12). This trade-off is not discussed.
- The method requires pre-segmented tasks with known counts ($|D_i|$), making it hard to apply directly to unsegmented "in-the-wild" datasets.

**Questions:**

- Q1.  Can you explain the significant HRT performance drop in Figure 4 (Pick/Place task)? Why did this trade-off only appear in the RoboCasa experiment and not in the Libero or real-world ones? Is this trade-off inherent?
- Q2.  The ReMix baseline performed very poorly (Fig. 4). Can you provide details on its tuning? How can we be sure this isn't just due to sub-optimal hyperparameters?
- Q3.  How can this method be applied to unsegmented "in-the-wild" datasets, given its reliance on known task counts ($\|D_i\|$)? Are there heuristic approaches (e.g., language clustering) that could work, beyond waiting for future skill segmentation research?

---

> ### Author Response · Authors · 2025-12-04
> **Reponse to Reviewer PbNn**
>
> We thank the reviewer for recognizing the importance of the action-primitive imbalance problem and for acknowledging the simplicity, efficiency, and comprehensive evaluation of our method.
>
> *Re: Weakness 1*: Please refer to Common Response #2.
>
> *Re: Weakness 2*: Please refer to Common Response #1.
>
> *Re: Q1*: Please refer to Common Response #2.
>
> *Re: Q2*: Please refer to Common Response #3.
>
> *Re: Q3*: Please refer to Common Response #1.

---

### Official Review · Reviewer_rWXB · 2025-10-30

**Soundness:** 2
**Presentation:** 2
**Contribution:** 2
**Rating:** 4
**Confidence:** 4

**Summary:**

The authors propose a sampling method for training robotic models under imbalance datasets. They propose temperature sampling, which increasingly up weights under represented tasks towards the end of training, and show that this helps the success rate of those under represented tasks.

**Strengths:**

- The method they propose is simple to understand and easy to implement compared to other works, and is shown to work well in practice in two simulated robotic environments and one real robot set up.
- The authors provide good ablation studies to show how the method works with different model size, different datasets, and different annealing process.

**Weaknesses:**

- It is unclear to me how the authors arrived at this particular form of temperature sampling. For example, instead of using |D|^(1/t), why not exp(|D|)^(1/t) for example? What about other forms? It would be nice to see how different forms for temperature sampling impacts policy performance.
- it would be nice to see whether this method works with larger and more realistic datasets. Currently, the authors hand pick a subset of Robocasa & LIBERO for their experiments. It is unclear in the paper how this subset and its composition was chosen, and how the results would differ when the dataset is chosen differently. On the other hand, it is much more natural to use a dataset like DROID, which is collected by human teleop and has natural data imbalance. Would training a VLA (e.g. OpenVLA, Pi0, etc.) over the DROID dataset, with temperature sampling, be better than vanilla random sampling?
- As the authors acknowledged, temperature sampling requires pre-specifying the number of steps to train. However, the optimal checkpoint for a different datamix may be different. For example, when we downweight HRT, we may need to train for longer to get good performance on HRT. The authors only compare on single checkpoint in the paper.
- lack of good baseline comparisons: the authors only compare against on baseline, ReMix, and it seems weird that it always perform worse than random sampling. Is there any reason why? The authors mention that ReMix is sensitive to hyperparameters: did you tune the hyperparameters when reporting its performance? If so, to what extent was the tunning effort? The paper would also benefit from additional baselines, such as [1]
- Temperature based sampling seems to significantly hurt performance on high resource tasks

[1] DataMIL: Selecting Data for Robot Imitation Learning with Datamodels

**Questions:**

- the authors mentioned that ReMix is sensitive to hyperparameters? Did you tune the hyperparameters when reporting its performance? If so, to what extent was the tunning effort? Why does it almost always perform worse than random sampling?
- Why do you need to use different base models for the Robocasa dataset vs the LIBERO dataset?
- How did you pick the checkpoint (40k)? Do you get similar results (i.e. ranking of different methods) if you pick other checkpoints?

---

> ### Author Response · Authors · 2025-12-04
> **Response to Reviewer rWXB**
>
> *Re: Weakness 1*: While we agree that there could be different formulations of the temperature sampling,
>
> *Re: Weakness 2*: Due to the resource requirements of training a large scale VLA model, we scoped our problem to train smaller models and fine tune larger scale models. Since fine-tuning larger models is a widely supported path in robot learning, showing these results in fine tuning is by itself significant. We are interested in investigating these results in larger scale training in future work.
>
> *Re: Weakness 3*: Please refer to Common Response #2.
>
> *Re: Weakness 4*: Please refer to Common Response #3.
>
> *Re: Weakness 5*: Please refer to Common Response #2.
>
> *Re: Question 1*: Answered in response to Weakness 4
>
> *Re: Question 2*: We picked the current state-of-the-art results for both of these dataset, and used those as our base-models for training/finetuning.
>
> *Re: Question 3*: We picked 48-hours GPU hours as our definition of fixed-compute and pick up the checkpoints that we obtain after 2-days of training.

---

### Official Review · Reviewer_rYJj · 2025-11-01

**Soundness:** 1
**Presentation:** 2
**Contribution:** 1
**Rating:** 2
**Confidence:** 4

**Summary:**

The authors explore the problem of choosing sub-dataset sampling frequencies when training imitation learning policies on imbalanced robot datasets. They study in particular this problem for cases where the imbalance arises due to the existence of different splits of the data, each split corresponding to a particular action primitive, with certain splits containing more data than others. This assumption is common in many large robot datasets where pick-and-place primitives are far more common than other types of robot skills, like wiping or throwing. The authors analyze two methods for choosing sampling frequencies based on two prior papers that studied this problem for multilingual LLMs: (1) upsampling low-resource data splits at the beginning of training and annealing it towards the end [1], and (2) increasing the sampling frequency of low-resource datasets towards the end of training [2]. The authors find that method (2) works better on a testbed RoboCasa simulation experiment, and on a real robot experiment. They also when proposing their method run it on a toy sparse parity task, where similar trends as in their main experiments are demonstrated. They also evaluate on the LIBERO simulation and report results of fine-tuning a foundation model.

[1] Li, Tianjian, et al. "Upsample or Upweight? Balanced Training on Heavily Imbalanced Datasets." arXiv preprint arXiv:2410.04579 (2024).

[2] Choi, Dami, et al. "Order matters in the presence of dataset imbalance for multilingual learning." Advances in Neural Information Processing Systems 36 (2023): 66902-66922.

**Strengths:**

(1) The problem tackled of picking sub-dataset sampling frequencies is an important problem in robotics, especially as more recent work ventures to train large robot policies on more heterogeneous data sources.

(2) The related works section of the paper is thorough

(3) The overall method makes sense, and experiments show better downstream task performance for their proposed sampling method in both real and sim.

(4) The authors compare against relevant baselines, including Re-Mix and the simple baseline of upsampling low-resource datasets and fixing this sampling ratio throughout training.

**Weaknesses:**

(1) The particular problem statement itself is not thoroughly justified. The problem statement in question is how to pick sampling frequencies when there exist subsets, each with different amounts of data for different action primitives. While action primitive decomposition is indeed one way of splitting data, it is not the only one, and there are many other axes upon which there exist low-resource and high-resource splits, such as language instructions, camera viewpoints, lighting, environment diversity, etc. By focusing just on action primitives, the authors missed the chance to study the robotic dataset sampling problem in its full scope.

(2) The authors mention related work on automatic skill decomposition via learned latent skills, but they do not actually run these methods (or any other method, e.g., based on parsing language instructions) to automatically segment robot datasets into skill-oriented subsets. Instead they assume that such decompositions already exist. However for many real world datasets such skill-oriented decompositions do not exist.

**Questions:**

(1) While I agree that pick-place tasks are overrepresented in large robot datasets, it would be good to include some evidence of this, e.g., by citing previous work.

(2) Section 5.2 is missing why the authors think the technique of upsampling low-resource datasets later in training works better than other methods.

(3) The authors justify choosing cosine warming after a hyper-parameter search and say it outperforms linear/exponential warmup/decay. The authors should include the actual ablation table or curves in the main text or appendix.

---

> ### Author Response · Authors · 2025-12-04
> **Response to Reviewer rYJj**
>
> *Re: Weakness 1*: Please refer to Common Response #1.
>
> *Re: Weakness 2*: Please refer to Common Response #1.
>
> *Re: Q1*: Thank you for raising the question, we have included citations to one of the previous papers where this imbalance is shown. [ See introduction in magenta]
>
> *Re: Q2*: Please refer to Common Response #4.
>
> *Re: Q3*: We have already included such metrics in our Ablation under Table 2.

---

### Author Response · Authors · 2025-12-04
**Common Response to Reviews**

We thank all reviewers for their thoughtful feedback. We are encouraged by comments such as "addresses an important and timely problem" (R1–R4), "simple and practical" (R2–R4), "comprehensive evaluation including real robots" (R2–R4), and "strong gains on low-resource tasks, more stable than ReMix" (R3).

**COMMON CONCERNS AND OUR RESPONSES**

1. *Scope of imbalance; reliance on segmented datasets (R1-Weakness 1, R3-Weakness 2, R4-Weakness 6)*
   *Concern that we focus only on action-primitive imbalance and assume pre-segmented data.*

   While we agree that there are many possible axes of a data-distribution that could be used to classify into high-resource and low-resource, we have explicitly made the focus on action-primitives since we believe that action primitives/skills forms an important basis of the distribution that is underexplored and largely ignored in most prior work studying data imbalance in multitask policy learning. There has been much prior work and methodological innovation on generalization particularly with respect to object classes and object geometry [1,2].
   [1] https://arxiv.org/abs/2310.14386
   [2] https://sites.google.com/view/hodor-corl24

2. *Trade-offs on high-resource tasks (R2-Weakness 5, R3-Weakness 1, R3-Q1, R4-Weakness 3)*
    *Figure 4 shows small degradation despite our "no degradation" phrasing.*

    We agree and will revise the claim. Our intent is to optimize average performance under fixed compute/model capacity, where modest HRT drops are an expected consequence of prioritizing underrepresented tasks. We will make this trade-off explicit.
   [See changes in magenta: Abstract (line 99), Introduction Contributions paragraph (line 122), Figure 4 caption (line 332), Figure 7 caption (line 396), and Summary paragraph (line 507)]

3. *Baseline coverage and ReMix performance (R2-Weakness 4, R3-Q2, R4-Weakness 2)*
   *Questions about weak ReMix results and missing baselines in real-world experiments.*

   ReMix requires training 3 models, and early stopping to decide weights for sampling. And we observed that timestep of early sampling can have huge variance in the final weights used for the sampling. We tuned the weights using 3-checkpoints, and reported the best results. We also contacted the authors of the paper for the best-practices of running the baseline to our particular setting. Real-world runs are constrained by robot time, so we focus on the most widely used baselines; extensive simulation comparisons complement these results. We will clarify this limitation.

4. *Limited explanation for why the warming schedule works (R1-Q2, R3-Q1)*
   *Section 5.2 lacks discussion of underlying mechanisms.*

   We will add hypotheses based on early shared representation learning followed by late-stage specialization, and a curriculum-like progression from data-rich to data-scarce tasks.
   [See Section 5.2, line 326 in magenta]

5. *Incremental novelty - Temperature-based sampling is known; how is this different? (R4-Weakness 1)*

   Response: Our study shows robotics benefits from the opposite schedule used in NLP (warming vs. decay). Combined with systematic evaluation across simulation, fine-tuning, and real robots—and the first thorough study of action-primitive imbalance—we believe the contribution is meaningful. We will strengthen this framing. [See Introduction in magenta]

6. *Low absolute performance in Figure 7 ablation (R4-Weakness 4)*

   The low Pick/Place performance in Figure 7 is due to extreme imbalance ratios (300:50 and 1000:50) and limited model capacity. These experiments were designed to stress-test the method under severe imbalance. Results are consistent with the original RoboCasa paper due to high task diversity (dozens of object categories with varying affordances) and dexterity requirements for Pick/Place tasks.
   [See ablation section line 379 in magenta]

We appreciate the reviewers' constructive insights and we have incorporated these clarifications to strengthen the paper.

---

### Meta-Review · Area_Chair_RgnS · 2026-01-05

**Summary:**

The paper studies how to train multitask robot imitation learning policies under data imbalance, focusing specifically on imbalance across known subsets or action primitives (e.g., pick-and-place vs. wiping). The authors propose a temperature-based sampling strategy with a cosine warming schedule, which progressively increases the sampling of low-resource tasks later in training. Through experiments on a toy problem, simulated benchmarks (RoboCasa, LIBERO), fine-tuning of a foundation model, and a real Franka robot, the authors demonstrate improved performance on low-resource tasks and higher average task success under fixed compute, compared to random sampling, fixed upsampling, and ReMix.

Reviewers are generally concerned about novelty, scope of the problem/assumptions of datasets, and fair comparison with baselines.

**Reviewer Concerns:**

Reviewers raised concerns about limited novelty, narrow scope, assumptions of segmented datasets, performance trade-offs on high-resource tasks, baselines, and insufficient explanation of why warming works.

The authors addressed concerns over paper presentation: explicitly corrected overclaims about “no degradation,” reframing the method as a compute–capacity trade-off; clarified ReMix tuning and real-world baseline limitations; added an intuitive curriculum-based explanation for the warming schedule; and contextualized low absolute performance in extreme imbalance ablations.

However, several concerns remain only partially or not addressed: the reliance on pre-segmented tasks is defended but not mitigated with practical alternatives; the method is still not validated on large, naturally imbalanced datasets; checkpoint dependence and fixed training-horizon assumptions are not resolved; and the specific temperature formulation is not justified or compared against alternatives.

Overall, the response improves clarity and framing but does not fully close the main gaps around generality, novelty, and robustness.

**Reviewer Scores:**

The author's response addressed clarity and overclaims, likely strengthens the two borderline reviews but is unlikely to convert the two rejects.

---

### Decision · Program_Chairs · 2026-01-26

Reject